# Contrasting Traditional and Virtual Teams within the Context of COVID-19 Pandemic: From Team Culture towards Objectives Achievement

**Mădălina-Elena Stratone** [1], **Elena-Mădălina Vătămănescu** [2,*] , **Laurențiu-Mihai Treapăt** [2] , **Mihaela Rusu** [3] **and Cristian-Mihai Vidu** [1]

1    Doctoral School in Management, National University of Political Studies and Public Administration (SNSPA), 30A Expozitiei Blvd., 012104 Bucharest, Romania; madalina.stratone@facultateademanagement.ro (M.-E.S.); cristian.vidu@facultateademanagement.ro (C.-M.V.)
2    Faculty of Management, National University of Political Studies and Public Administration (SNSPA), 30A Expozitiei Blvd., 012104 Bucharest, Romania; laurentiu.treapat@facultateademanagement.ro
3    Doctoral School in Communication Sciences, National University of Political Studies and Public Administration (SNSPA), 30A Expozitiei Blvd., 012104 Bucharest, Romania; michaelarusu@yahoo.com
*    Correspondence: madalina.vatamanescu@facultateademanagement.ro; Tel.: +40-7481-19900

**Abstract:** The current paper aimed to comparatively scrutinize some key dimensions apposite for the dynamics of traditional versus virtual teams in the context of the COVID-19 pandemic. Emphasis was laid on the positive leadership perception, communication effectiveness among team members, objectives achievement, diversity approach, and the perception of team culture strength. Invitations to fill in an online questionnaire consisting of paired items were sent in January 2022 via email to over 200 potential respondents working in both traditional and virtual teams, using a snowball sampling technique; 137 subjects completed the entire questionnaire, hence allowing a reasonable research sample for conducting relevant statistical analyses (i.e., paired *t*-test given that the aim was to investigate the difference between paired sets of variables for the same issues). The empirical exploration brought to the fore significant differences among the considered dimensions, thus underscoring the benefits and drawbacks of working in traditional versus virtual teams in the context of the new normal. Evidence was brought forward that teamwork in traditional teams (i.e., based on face-to-face interaction) is preferable to that in virtual teams (i.e., based on online interaction). Meaningful differences were observed regarding the perception of team culture strength, communication effectiveness, positive leadership perception, and diversity approach in favor of traditional teams, the questioned respondents opting for the offline coordination and collaboration processes.

**Keywords:** traditional teams; virtual teams; COVID-19 pandemic; team culture; leadership; communication; diversity; objectives achievement

## 1. Introduction

The COVID-19 pandemic has emerged as a multi-level crisis, directly afflicting most of the subunits of the ecosystem, from the individual agent towards the global society as a whole. People, communities, organizations, authorities, societies, and the labor market were found totally unprepared to deal with the new surge and disposed of no meaningful resources to take prompt action [1–7]. Among all social actors, organizations were dared to find suitable ways to survive and pursue their socio-economic goals in a context where social isolation became the norm and therefore the first measures to be applied envisaged the development of hybrid work environments [4]. These environments (integrating face-to-face and online interactions among organizational members) were perceived as a solution at hand for a viable and sustainable adaption to the new normal.

Over the last years, the working relations left the physical offices and cooperation in a virtual environment, using the most up-to-date technologies, has exponentially increased.

As Plavčan and Funta [8] contend, the internet platforms have advanced rapidly and complexly, affecting people and businesses and reconfiguring how information is exchanged and centralized and how technological disruptions impact society as a whole. In line with Canary and McPhee [9], once the time and space limitation has been cancelled, the access to a comprehensive pool of knowledge via electronic networks becomes more and more facile and effective. Despite the fact that many leaders have acknowledged these trends alongside the disruptions caused by the COVID-19 pandemic, more substantial changes are expected in order to meet the challenges of hybrid work [10–12]. For instance, the dynamics of communication flows within the company, of how tasks can be efficiently handled and job satisfaction still attained in an organizational context where traditional teams and virtual teams cohabitate is yet to be properly explored [3,13,14].

Generally speaking, virtual teams may look quite similar to traditional teams, but the reality shows that a lot of differences are to be noticed and, at the same time, new and unexpected challenges may occur while working with or inside a virtual team. According to Wheelan [15], we can count some similar aspects when it comes to the features of a high-performance team, such as a good understanding of the common goals among the team members, assigning appropriate tasks for each member, a comprehensive understanding of the roles and duties inside the team, including all the team members in the day-to-day activities by using proper communications methods, and a structured approach, allocating time to discuss, define, and fix the eventual problems and bottlenecks as a group, applying strategies that are effective when it comes to decision-making inside the team, welcoming subgroups inside the team as a whole. Moreover, in order to be productive, the teams have to count a limited number of members, strictly adjusted to the targeted goals, as working with larger teams is not always an advantage, as they tend to divide into small-sized "social-psychological groups who see themselves attached more to one another in smaller informal groups to which they have been assigned" [16] (p. 284). In addition to these, a team has to be characterized by a strong degree of cohesion and a good cooperation inside and with the third parties involved, it being necessary for the members and partners to spend enough time together so as to develop a consistent and mature working environment and manner and a good conflict management policy while being capable of keeping focus on the objectives and targets, showed Wheelan [15]. All these features lead to the conclusion that "10 key areas members should pay attention to in order to ensure the productivity of the group: goals; roles; interdependence; leadership; communication and feedback; discussion, decision-making and planning; implementation and evaluation; norms and individual differences; structure; and cooperation and conflict management", as explained by Wheelan [15] (p. 60).

Assuming these issues, the literature approaching the topic of virtual teams is currently developing, thus enhancing the volume of knowledge about their challenges and benefits, their bottlenecks, and dynamics [16–18]. As the COVID-19 pandemic has transformed the issue of traditional versus virtual teams into a moving target, many aspects are still not fully covered, hence triggering the imperative to continue the research in the field and to contrast different types of work collaboration [19]. In this front, Brătianu and Bejinaru [3] contend that staying home and working from home also created a new way of doing things, communicating and interacting, altering the patterns of knowledge sharing and of team spirit. Brower [20] believes that leadership-related issues will be challenged during crisis, organizational culture will be brought to the foreword, and working from home will be the new norm alongside more frequent team engagement, vast flexibility, and significant use of technology. As revealed by EY [21], employees are nowadays striving for more flexibility and a greater mix of work from home in the future. They want to return to the office for social contact and are looking to their employers to enhance digital tools for virtual working.

Taking into consideration this significant mental shift apposite for employees, the new trends in the workforce's preferences, and the substantive pressure of leaders to attract and keep highly skilled and talented employees motivated and satisfied, there is a pivotal

exigence to further examine the significant differences between relevant organizational processes taking part in virtual teams (i.e., interacting and working online) and traditional teams (i.e., interacting and working face-to-face). To this end, the current paper intended to comparatively scrutinize some key dimensions, namely positive leadership perception, communication among team members, objectives achievement, diversity approach, and team culture. Invitations to fill in a research instrument consisting of paired items were sent in January 2022 via email to over 200 potential respondents working in both traditional and virtual teams, using a snowball sampling technique. Of these, 137 subjects completed the entire questionnaire, hence allowing a reasonable research sample for conducting relevant statistical analyses. The empirical exploration brought to the fore significant differences among the considered dimensions, thus underscoring the benefits or drawbacks of working in traditional and virtual teams.

By unfolding the present empirical investigation, the paper sets out to contribute to the extant literature in two main ways. Firstly, to the best of our knowledge, this is the first undertaking to comparatively scrutinize traditional and virtual teams at different levels of communication and interaction in the context of COVID-19 pandemic. As further elaborated on in the literature review section, various studies [10–13,16,17] have addressed team dynamics from distinctive standpoints in the past, yet a research gap still remains in terms of a topical and more comprehensive comparison between the two types of teams within the framework of a systemic crisis that affected all organizational layers and especially the human capital [3,21]. "The evolving practical problems in business practice" [22] (p. 454) call for future analyses able to make a step forward into the understanding of the processes inherent to team communication, culture, leadership, and achievement.

Secondly, according to the United Nations [23], the pandemic has led to the loss of almost 255 million full-time jobs whereas economic and social recovery is under way. In this front, one of the most important sustainable development goals is the promotion of full and productive employment and decent work for everybody given the impact exerted by the virus outspread on society, labor, organizations, and their human capital. Subsequently, exploring key variables of teamwork in the context of the new normal avails an articulate perspective on a pivotal component of the Triple-Bottom-Line, namely people [4,24]. A multifaceted outlook of people well-being and accommodation to novel and challenging working conditions is brought forward in an effort to better comprehend which of the two types of teams is more beneficial from a sustainability perspective. From a sustainable development approach, the main questions to be answered are which of the traditional and virtual teams succeeds in cementing sustainable organizational structures via the resolution of multi-level group and social issues, the consolidation of a participatory environment and of a strong culture, and the promotion of a trust-based and engaging work climate which supports accountability, a sense of responsibility, and achievement [25]. All these questions are addressed through a stepwise theoretical and empirical appraisal which intends to dialectically, yet integratively discuss the perception of team members on communication, leadership, culture, diversity, and objectives achievement.

With a view to unfold a round-off perspective, the paper is organized as follows: firstly, the literature review and the hypotheses development are introduced, and secondly the research design is presented. Thirdly, the results and their discussion are related. The last section covers the conclusions, implications, limitations, and future research avenues.

## 2. Literature Review and Hypotheses Development

The social distance imposed by the COVID-19 pandemic alongside the development of technology has positively affected people's personal lives and the every-day working manner and environment. Currently, people can communicate and work together remotely, from different locations, cities, or even countries, very similar to the way they would work while sharing the same office. The world displays a brand-new face we have never seen before, while the business world encounters challenges never experienced until now (i.e., the imperative to reimagine work conditions in order to create a productive climate for

all parties involved via a mixture of face-to-face and virtual interaction) [10–12,18]. As the result of the global virtualization trend, new types of organizations have emerged, being composed of work groups aiming to promote innovation and that have the purpose to increase the capacity of work [26], and the phenomenon has had an unexpected rhythm since the outbreak of the pandemic [19].

The digitalization of the work processes has enabled remote work and the building of virtual teams in various fields of activity and for all types of companies and entities. Thus, the virtual society started to expand the global electronic space and information became more and more available through technology. Getting in contact with coworkers and leadership and sharing work and results remotely is no longer a challenge, but business as usual [13]. The term "virtuality" has a large range of meanings in the specialized literature. The global dimension of virtualization is permanently compared to the features of the local and regional traditional teams, when it comes to the dimensional and relational aspects, involving geographical dispersion and electronic communication and interconnection [17,27]. New typologies of organizational structures and the new forms of organizations emerged (i.e., organizational structures availing more flexibility in terms of schedule, location, co-presence), most of them being a virtual response to a present, more complex business environment, impacted by turmoil and uncertainty, thus creating new ways and means for both people and organizations worldwide to benefit from fresh opportunities (i.e., a better work-life balance, intercultural and transnational teamwork and coordination, etc.) [28–31].

## 2.1. The Perceptions of Team Culture Strength in Traditional versus Virtual Teams

Unlike social groups where the members can influence each other's work during their normal interaction, teams are defined as more specialized groups, having a strong ownership and commitment that builds the collective identity in its whole [32–34]. Other theorists describe how the behavior of the teams is different from the behavior of a group, defining the teams as groups of people with a certain structure, working to achieve targeted common goals in a coordinated interaction [35]. The cooperation among the members of both traditional and virtual teams is characterized by the dependency everyone is experiencing in relation with their colleagues on the assumed methodology to accomplish the assigned work and achieve the targets [17,36]. In this vein, some communicational and interactional patterns and processes come forward as robust indicators of team culture, leadership perception, diversity management, and objectives achievement.

Pauleen and Yoong [37] focused on various papers analyzing the aspects of building relationships in the context of using information and communication technologies (ICT), and underlined that a conclusive challenge is to work at linking the individual culture of each team member into the team culture. Thomson [38] described the culture of a team as being a "reflection of the organization's culture"; thus, it is very important to develop the culture of a team, regardless of whether it is similar to the organizational culture or different from it, as a common culture unites the members of the team, conglomerates the working cell, develops a feeling of belonging, and enhances morale by increasing acceptance, tolerance, and understanding towards and among all the members. The main asset resulting from all these efforts is accepting the diversity in its entirety, an aspect of the highest importance for the team and for the targeted results.

The members of a work team, especially of virtual ones, must constantly redefine their positions, re-arrange the roles, and adjust their relationships and ways of collaborating with each other in order to negotiate a common ground and agree upon the tasks' boundaries, responsibilities, and personal duties [39]. Fostering a team culture is very important for team viability, and in the absence of a common history, difficulties may occur in assigning tasks and responsibilities, planning team development, and locating expertise, difficulties that can be overcome with trust, commitment, and good interpersonal and working relations among the members [13,17,19]. Rules become common values, trust is more solid, the relationships among members become stronger, and they start to rely on each other and accept challenges, steadily reifying team cohesion [40].



The team leader who is coordinating the people involved must be skilled, trained, and experienced in facilitating communication and cultivating a feeling of belonging. They have to promote and develop values like cooperation, support, mutual understanding, and even empathy and tolerance when the pressure inside the team reaches high peaks, before deadlines, or in difficult moments [18,41]. All these have to build a certain degree of confidence and trust among the members of the respective team, as Nemiro et al. [42] showed, as the best way to develop "a common team identity is to increase team members' confidence in each other and the team as a whole", to enhance the feeling of "common membership" [43] (p. 93). Team leaders should also attach the proper attention to sensitive topics such as anti-discrimination and wages; working in virtual teams, with members from different countries, requires awareness of the legislation, understanding concepts (such as equal treatment, direct and indirect discrimination, equal pay for work of the same value, remuneration of each member), and the ability to handle the cultural barriers by increasing awareness on the anti-discrimination policy in labor law legislation and on cultural diversity and its benefits [6,7,22].

Regardless of whether we speak about traditional teams or virtual teams, each member has to fully understand the roles, tasks, and targets that are assigned to them for the team to be effective and run in good conditions. This could be a difficult task to achieve sometimes as, particularly for the case of the virtual times, face-to-face meetings would be recommended at least at the beginning of a project but given the space and time limitations, doubled by the budget restrictions, this is not always possible. As Fisher and Fisher [44] pointed out, "creating a team culture that is supportive and productive is especially helpful in teams with only minimal face contact", a statement that highlights the real importance of face-to-face kick-off meetings. However, even if working with virtual teams requires more management and coordination efforts given the reasons previously explained, a good and efficient start can overcome such disadvantages and difficulties [18,45,46]. By conflating the aforementioned arguments, we thus posit that:

**Hypothesis 1 (H1).** *There is a significant difference between the perceptions of team culture strength in traditional versus virtual teams.*

### 2.2. Positive Leadership Perceptions in Traditional versus Virtual Teams

The individual profile, background, education, skills, and experience of the team members can be decisive when starting a virtual team. For efficiently managing such situations, choosing a qualified team leader is a must, because only a person having previous experience in working with people can handle existing differences and eventually is able to turn them into advantages and opportunities when creating an identity and a common culture of the team involved in a project they manage [41,46]. Moreover, understanding the cultural background of each person inside a team will "enable any leader to more effectively address communication and behavioral differences that arise in virtual teams" [47] (p. 277), generating trust and confidence among the members. In line with Jordan and Adams [48] (p. 2), any kind of diversity, be it disciplinary, educational, cultural, or even if it refers to geographical areas or to the individuals' ethnicity, age, or gender, it "serves to maximize the number of different viewpoints, approaches and frames of mind".

Kayworth and Leidner [49] stated that effective leadership basically translates the perception the members of the team have regarding the effectiveness of communication, in how satisfied they are about the way the communication is performed inside the team, and in the team leader's ability to define and assign proper roles and responsibilities. Fisher and Fisher [44] clearly listed distinct roles a team leader should assume when coordinating a team: to be a living example, a role model for his subordinates, to be a coach, ready to help the other members to improve their working skills, abilities, and the necessary competencies according to the requirements of the activities they are performing, to be a good business

analyzer, able to turn the chances and challenges into opportunities, to be a barrier buster, by running interference for the team, to be a facilitator and offer all the necessary tools, resources, and information to the team members in order for them to successfully provide the required deliverables of the project, and to be a results catalyst, helping his team members in their work for achieving the expected results and for continuously improving their work performance. Further, Duarte and Snyder [50] pointed to four competences which are critical for coordinating virtual teams in an effective and efficient manner and, here, the authors mentioned communication, establishing the expectations and resources allocation, and modeling the required behaviors inside the team.

Compared with the dynamics apposite for the traditional teams, in virtual teams, changes happen more frequently while negatively impacting the processes running inside the team [27]. Working in a virtual team involves more dynamic and flexible forms of organizing the activities such as alliances, outsourcings, offshoring, and temporary project-based work. The many changes that might normally appear in a virtual team attract a lot of vulnerabilities which call for a thorough leadership approach [13,18,41]. A team leader should acknowledge such inconveniences to mitigate them in due time, focusing on building "relationships among team members", and also on implementing and developing "team processes" [51] (p. 3). Additionally, team leaders have to prove their abilities to manage interaction and communicational issues, to support individuals, and to keep the team together by being open to discussions and to problems solving and by providing a clear picture of the common objectives and goals [41,52,53].

In traditional teams, where the interaction happens face-to-face, leadership is a key aspect that influences the individuals' attitudes and behaviors and at the "team level impacts not only team processes and outcomes but also individual effectiveness" [54] (p. 3). On the other hand, as Hoch and Kozlowski [55] underlined, at the entire team's level, in case of the virtual teams, leaders are involved in more than just creating, developing, and designing the processes within: they are also in charge with managing and monitoring the performance of their teams. For creating an effective and efficient virtual team, "both leaders and members of virtual teams, even if experienced with face-to-face teams, need enhanced competencies to be effective" [56] (p. 17). Moreover, the team leaders coordinating virtual teams have to perform all the necessary tasks to "create, reinforce, and maintain trust between the members of their teams as well as between themselves and their team members" [51] (p. 3), as the basic value of a team must be the trust, in the absence of which nothing would be achievable. The leaders of the virtual teams have a different activity compared with the ones of the traditional teams. They have to play both roles, i.e., of team leader and of team member, and for this reason they must have specific competencies and abilities required by the remote way of working, namely technological proficiency, intercultural skills, remote leaderships skills, etc. [19,45]. Based on these facts, we propose the following hypothesis:

**Hypothesis 2 (H2).** *There is a significant difference between positive leadership perceptions in traditional versus virtual teams.*

### 2.3. Team Diversity Approaches in Traditional versus Virtual Teams

Nowadays, a main feature of many work teams is diversity, as the members may come from different regions and speak different languages. Additionally, the cultural diversity, the customs, and the values that depict a virtual team in its entirety are to be seen as pillars and, at the same time, approached somehow as integrative factors in a heterogeneous group of people, as Oliveira and Scherbaum [57] also explained. Consequently, these aspects have to be managed accordingly, so as to increase the potential and the performance of the teams, be they traditional or virtual. This is a very important aspect to be considered as, in a high diversity environment, communicational barriers and all sort of conflicts may occur, even though a common language is used for communicating at work.

To really create a team that functions as one body and achieves the best results in terms of efficiency and effectiveness, the team leader must create an identity for the team as well as a common culture and the feeling of belonging. In this regard, identity is one of the main pillars and plays "a crucial role in communication because knowledge of those with whom one works and communicates is necessary for understanding and interpreting interaction styles", according to Beyerlein [45] (p. 48).

Nowadays, people's perceptions towards different cultures have changed and evolved, currently showing a better understanding and openness to diversity. Virtual teams imply working with people from all over the world, which belong to different environments and have personal values deriving from their cultures and traditions, and also different ways of working and approaches [13]. In this context, attention to detail is essential and consequently, the team leader has to find efficient ways and means to accommodate the members of the team with each other and inside the virtual team, as a whole. This is a very important issue as only in this way can further conflicts, misunderstandings, or stereotypes can be avoided later on [47]. The language used is very important for a good and comprehensive communication and understanding; slang words and street language should be avoided in all cases, and common words are preferable to be used for understanding the assigned tasks fully and correctly [58].

Misunderstandings in terms of values, approaches, and communication errors may occur, negatively affecting the results and the performance of the respective team [41,52,59]. It is thus advisable to encourage values alignment and reconciliation to grant the comfort and the pride for its members and the feeling that they belong here, and that they can find trust and mutual support. As Levi [47] showed, if handled improperly, diversity can ruin the cohesion and communication at all levels inside the team, as the members may develop connections with one another based on the similarities they share and prefer to interact only with similar peers to a certain extent. In this respect, face-to-face meetings taking place within traditional teams proved to be major prerequisites for strengthening the bounds that keep the team together [60]. Building on these arguments, we advance the following hypothesis:

**Hypothesis 3 (H3).** *There is a significant difference between team diversity approaches in traditional versus virtual teams.*

### 2.4. Communication Effectiveness in Traditional versus Virtual Teams

Good communication is a prerequisite of any successful relationship and has to be accomplished by using a certain channel and a common language, unanimously accepted by all the participants, as, in order to build an efficient communication, the transmitted message has to be decoded and understood by both the issuer and the recipient [61,62]. Pursuant to Levi [47], in the case of the virtual teams, communication represents one of the core activities, facilitating cooperation, triggering new ideas, and stimulating creativity and the out-of-the-box thinking among the members. Working and communicating together in an efficient manner, the team will be able to adapt "to the changes in the organizational culture, because it encourages adequate participation of workers" [63] (p. 5) and to efficiently achieve the targeted goals and objectives.

In the case of the virtual teams, the technology of information and communication is a must for an efficient cooperation and coordination of the common efforts in accomplishing the targeted goals [13,18,27,64]. Team dispersion directly influences the dynamics of the communication among team members (i.e., co-workers) and imprints a different trademark to the coordination, clarification, and edification processes [65,66]. Gibbs et al. [67] (p. 4) proposed arguments in favor of computer-mediated communication, positing that "the degree of information value (e.g., communication richness) and synchronicity of communication technology" should be approached as "key elements of virtuality". Likewise, as Purvanova [68] explained, if the team members succeed in increasing the exchange of social

information, technology can be turned into an opportunity. Yet, remote communication is less personal and does not provide the opportunity for the members of the same team to get to know and to understand one another better, to create stronger bonds and relations, but it also can prevent the eventual preconceptions towards the physical features or the ways of behaving some colleagues might have [19,69]. In other words, remote communication is mostly soulless and objective while face-to-face communication might involve personal bonds like affinities and friendships in addition to work cooperation, facts that might lead to subjective approaches inside a team.

Krawczyk-Bryłka [59] (p. 2) showed that the language barriers can represent a real issue, resulting in communication breakdowns and, possibly, in a lack of self-confidence, as "language barriers intensify isolation and frustration". Consequently, tasks must be assigned also considering this aspect for enabling fruitful cooperation among peers and diminishing the negative impact of the language barrier. Klitmøller et al. [26] posited that, when choosing communication tools, the decision makers have to keep in mind the language skills of each member. If we deal with less proficient people, then textual communication is preferable, giving the possibility to re-read, re-think, and correct possible mistakes [26]. Language skills are very important when working in a virtual team, as, in the absence of good linguistic competencies, the effectiveness and the efficiency of the entire team will suffer on one hand, while on the other hand, it will negatively affect the communication and the trust that has to be permanently cultivated as, according to Child [70], building trust can reduce the cultural differences and distance that can occur in any virtual team. Trust can enhance the motivation and facilitate a better flow of information, thereby increasing the performance and avoiding and fixing the possible conflicts that might normally occur when many people work together [70]. Communication issues might occur especially in large-sized teams as we have explained before, and thus subgroups might be formed, in most of cases affecting the cohesion of the team as a whole [43]. On the background of this lack of communication that can generate misunderstandings when it comes to tasks and responsibilities, and particularly when the team does not have a well-formed identity, interpersonal conflicts are also likely to appear.

In contrast, for the traditional teams, technology represents just a tool that supports face-to-face work, for example, when delivering presentations during meetings or working on shared projects, and also in electronic communication and so on. In this way, the members of such teams could share their work and the necessary data and information and exchange views and opinions as an additional channel to the face-to-face interactions. Nevertheless, direct communication remains the backbone of an efficient and effective cooperation within the traditional teams, whereas online communication has become instrumental in the virtual teams [71,72]. Based on these arguments, we presume that:

**Hypothesis 4 (H4).** *There is a significant difference between communication effectiveness in traditional versus virtual teams.*

### 2.5. Objectives Achievement in Traditional versus Virtual Teams

Beyerlein and Harris [47] (p. 40) defined collaboration as "the collective work of two or more individuals where the work is undertaken with a sense of shared purpose and direction that is attentive, responsive and adaptive to the environment". Collaboration is the main means of accomplishing something together, to make a change, to progress, and to benefit from extended human capital acting in a joint way to achieve a common goal, as it is well known that the teamwork makes a difference when it comes to synergy, potential, effectiveness, and efficiency.

Teams represent groups of individuals who are expected to perform relevant tasks by sharing common goals, socially interacting, and by establishing task interdependencies [73]. Wageman, Gardner, and Mortensen [74] described teams as a "bounded and stable set of individuals interdependent for a common purpose" (p. 303). In this light, according to

Dunham [75], the affiliation to a group of individuals who are gaining experience on how to cope, interact, and work together, regardless of the type of activity, project, or the entity they are activating within, creates the prerequisites for growing personal success and collective survival and also enhances objectives achievement and organizational sustainability [75]. In this front, Wu and Cormican [76] brought to attention the responsibilities a team leader must take in relation with the team they coordinate, namely to organize their team in accordance with the developed activity and the targeted results, to identify and articulate clear goals and objectives, to assign tasks, duties, and responsibilities to suitable members, to follow-up the plan and make all the necessary adjustments as necessary and inform the others accordingly, to assess the performance of the members part of their team and provide feedback, to manage and allocate the necessary resources for running the project in good conditions, to facilitate the exchange of data and information, to encourage the members to grant support to one another, to involve themselves in solving the issues and conflicts that might occur during day-to-day activities, and to structure and conduct effective and efficient teamwork.

There are similarities when it comes to the problems that both the traditional and virtual team happen to face. Scholars indicated that among the five common problems that people are usually experiencing when working in a team, we can mention the lack of commitment, productivity losses, poor communication, interpersonal conflict, and poor leadership [75]. Thus, we can speak about a lack of commitment when only a few members of the team are fully involved in the work, the rest not following the pace of the others and displaying a detached attitude. In such cases, the productivity decreases, particularly when this unwanted situation overlaps with a poor structure, bad planning, or a weak decision-making process or, when there are all sort of conflicts and misunderstandings about the members' responsibilities, the team's objectives and deliverables. A reasonable solution would be defining clear roles and tasks inside the team, building trust among the members, peers, or leaders, and encouraging efficient and transparent communication at all levels [77].

Hoch and Kozlowski [55] affirmed that the shared leadership is a style of leading according to which the tasks and responsibilities are shared among the members of one team, regardless of whether it is traditional or virtual, in this way exerting a mutual influence upon one another and promoting a collaborative decision-making process. Na Chen [78] explained that, through a shared leadership, the feeling of equality and equity is preserved inside the team, particularly when we speak about virtual teams, so it can prove itself to be the most favorable leadership model, though a central figure must still exist, a focal point to whom the rest of the members still have to report to: a team leader, namely "the person who is managing the boundary, feeding the team's accomplishments to the organization and to the individuals' function or line managers" [79] (p. 5). In addition to the necessary resources that are granted to the team for running in the day-to-day work in good conditions, the members must also "have the desire and willingness to coordinate their efforts to work collaboratively" [44] (p. 183).

Nevertheless, when working remotely, it is not always easy to support, control, stimulate, or motivate the team, aspects that might turn into real challenges for the project. According to Jones et al. [69], there are several practices the team leaders can make use of, such as good and frequent communication through which colleagues can be informed and updated about the new priorities, challenges, and the eventual changes in the organization they are working for, fairness and openness towards their subordinates while avoiding hurting their feelings and egos, transmitting clear, direct messages and giving feedback using a friendly way of speaking and, not least, availability and accessibility, namely the team leaders to always be there for their people, whenever they are needed [69]. Conflating these issues, we thus posit that:

**Hypothesis 5 (H5).** *There is a significant difference between objectives achievement in traditional versus virtual teams.*

All in all, the review of the aforementioned literature suggested that previous studies have addressed different facets of traditional and virtual teams via a wide array of research designs and standpoints, yet no recent work has comparatively tackled the five dimensions under scrutiny in an integrative frame of discussion. Consequently, the current empirical analysis contributes directly to the extant body of knowledge by extending its scope and particularizing its focus on the new normal entailed by the COVID-19 pandemic.

## 3. Materials and Methods

### 3.1. Sample and Data Collection

Invitations to fill in the research instrument were sent in January 2022 via email to over 200 potential respondents, using a snowball sampling technique, as is usually applied in social sciences [80]; 137 subjects completed the entire questionnaire, thus yielding a response rate of 68.5%. The preliminary condition for filling in the online questionnaire was that all participants of the study had previous work experience in both traditional teams (i.e., team members that have face-to-face interactions) and virtual teams (i.e., team members that have online interactions). The sample (Table 1) comprised the answers of 137 respondents and can be characterized by heterogeneity in terms of gender, age, work departments (including: education public administration, human resources, management, marketing, baking, sales, and industry), and the type of organizations they are working in.

**Table 1.** Sample characteristics.

|  | **Frequency** | **Percentage** |
|---|---|---|
| No. of participants | 137 | 100% |
| Gender | | |
| Women | 81 | 59.12% |
| Men | 56 | 40.88% |
| Age | | |
| 18–24 | 31 | 22.62% |
| 25–34 | 61 | 44.56% |
| 35–44 | 30 | 21.88% |
| >45 | 15 | 10.94% |
| Type of organization respondents are working in | | |
| Small and medium-sized enterprises (SMEs) | 41 | 29.92% |
| Multinational corporation (MNC) | 46 | 33.58% |
| Public Organizations | 35 | 25.55% |
| Other types of organizations | 15 | 10.95% |
| Work Experience (Years) | | |
| 0–1 | 16 | 11.68% |
| 1–3 | 29 | 21.17% |
| 3–5 | 18 | 13.14% |
| 5–10 | 27 | 19.71% |
| >10 | 47 | 34.30% |

### 3.2. Method

The empirical data collected via the questionnaire-based online survey were statistically processed using SPSS version 26. In order to test the inferred relationship among

the variables, we used the paired *t*-test given that we aimed to investigate the difference between paired sets of variables for the same issues (e.g., communicational or interactional patterns within virtual versus traditional teams). Given the fact that the aim of the investigation was an exploratory one, the paired *t*-test rose as a pertinent method to test whether the mean difference between pairs of measurements is zero or not and consequently to confirm the statistical significance of the extant difference. The performed analysis was able to provide a preliminary synopsis of the state-of-the-art as a premise for more in-depth appraisals regarding the nature of the relationships among constructs which would rely on structural equation modelling techniques [81].

*3.3. Measures*

The questionnaire comprised 64 items, among which 46 statements were designed as paired items which addressed traditional versus virtual teams related issues. Ten main constructs were established, five for traditional teams and five for virtual teams. The presentation of the constructs (i.e., variables) and their items are illustrated in Table 2. All items were measured on a five-point Likert scale, ranging from 1 (To a very small extent) to 5 (To a very great extent). The items in the research instrument were adapted from previous studies, e.g., [13,18,27,38,39,41,44,47,64–68] addressing the same or similar issues, as specified in Table 3.

**Table 2.** Variables and items.

| Variables | Cronbach's Alpha ($\alpha$) | Paired Items * | Sources (Adapted From) |
|---|---|---|---|
| Perceptions on team culture strength in traditional teams (five items) / Perceptions on team culture strength in virtual teams (five items) | CULT_TRAD ($\alpha$ = 0.808) / CULT_VIR ($\alpha$ = 0.840) | 1. While working in Traditional/Virtual Teams, I feel that the needs of the team members are taken into consideration. 2. While I was a member in a Traditional/Virtual Team, I was surrounded by people who shared my values. 3. Team spirit can be easily developed within a Traditional/Virtual Team. 4. While working in Traditional/Virtual Teams, members are aware of the goals and objectives of the organization. 5. It is very important to have face-to-face/online meetings in order to create the culture of a team. | [13,38,39,41,44] |
| Positive leadership perceptions in traditional teams (four items) / Positive leadership perceptions in virtual teams (four items) | LEAD_TRAD ($\alpha$ = 0.846) / LEAD_VIR ($\alpha$ = 0.802) | 1. The Team Leader succeeds in supporting teamwork efficiently within a Traditional/Virtual Team. 2. Communication on tasks between the Team Leader and the members of the team is efficient within a Traditional/Virtual Team. 3. While working in Traditional/Virtual Teams, the Team Leader plays an important role, directly inspiring team members. 4. Face-to-face/online communication with the leader allows a better clarification of vague issues. | [18,41,47,49–51,53] |

**Table 2.** *Cont.*

| Variables | Cronbach's Alpha (α) | Paired Items * | Sources (Adapted From) |
|---|---|---|---|
| Diversity approaches in traditional teams (three items) / Diversity approaches in virtual teams (three items) | DIV_TRAD (α = 0.830) / DIV_VIR (α = 0.866) | 1. While working in Traditional/Virtual Teams, members diversity can be handled easier. 2. In face-to-face/online interactions, intercultural differences can be surpassed easier. 3. Values alignment is easier to achieve in Traditional/Virtual Teams. | [45,56–58] |
| Communication effectiveness in traditional teams (six items) / Communication effectiveness in virtual teams (six items) | COMM_TRAD (α = 0.896) / COMM_VIR (α = 0.903) | 1. The members of a Traditional/Virtual Team work together efficiently for common achievements. 2. The members of the Traditional/Virtual team communicate efficiently for problem solving. 3. The members of a Traditional/Virtual Team collaborate easier for tasks completion. 4. The members of a Traditional/Virtual Team share the responsibilities for tasks easily. 5. Face-to-face/online communication among the team members catalyzes a better understanding of how objectives should be approached. 6. Oral/online communication with co-workers facilitates the clarity of task completion. | [13,18,27,47,64–68] |
| Objectives achievement in traditional teams (five items) / Objectives achievement in virtual teams (five items) | OBJ_TRAD (α = 0.837) / OBJ_VIR (α = 0.877) | 1. The team offline/online meetings are conducted efficiently. 2. The members of a Traditional/Virtual Team can overcome the changes that they might face during a project easily. 3. The Traditional/Virtual Team meets efficiently the objectives of a certain project. 4. While working in Traditional/Virtual teams, the members finish the project tasks on time. 5. Working in a Traditional/Virtual Team is less time-consuming. | [74–79] |

* Separated items were formulated for the traditional and virtual teams. The items are compressed in the table for formatting reasons.

**Table 3.** Descriptive statistics.

| | | Mean | N | Std. Deviation | Std. Error Mean |
|---|---|---|---|---|---|
| | | **Paired Samples Statistics** | | | |
| Pair 1 | CULT_TRAD | 4.0584 | 137 | 0.74049 | 0.06326 |
| | CULT_VIR | 3.5255 | 137 | 0.87448 | 0.07471 |
| Pair 2 | LEAD_TRAD | 4.1296 | 137 | 0.80217 | 0.06853 |
| | LEAD_VIR | 3.5091 | 137 | 0.85610 | 0.07314 |
| Pair 3 | DIV_TRAD | 3.7810 | 137 | 0.92570 | 0.07909 |
| | DIV_VIR | 3.3698 | 137 | 1.03349 | 0.08830 |
| Pair 4 | COMM_TRAD | 4.0730 | 137 | 0.77197 | 0.06595 |
| | COMM_VIR | 3.4550 | 137 | 0.88573 | 0.07567 |
| Pair 5 | OBJ_TRAD | 3.6117 | 137 | 0.84497 | 0.07219 |
| | OBJ_VIR | 3.6000 | 137 | 0.83912 | 0.07169 |
| | SAT_VIR | 3.3406 | 137 | 1.17536 | 0.10042 |

### 3.4. Reliability and Validity Analyses

In order to pertinently assess the appropriateness of the employed constructs, reliability and validity checks were performed for all variables. For the reliability analysis, Cronbach's alpha ($\alpha$) was used. Each scale (i.e., construct) reliability was assessed using the specific predefined function of SPSS version 26. At this level, all the considered variables reported values above the threshold of 0.7, in line with Nunnaly [82], as seen in Table 2.

The validity analysis relied on the Pearson Correlation test, which appraised the correlation between each item in the questionnaire and its total value. Given the size of the sample (N = 137), the critical value for Pearson's Correlation Coefficient for the significance level ($\alpha$) < 0.05 was 0.168. As all the Pearson Correlation values of the questionnaire items with their sums were above this threshold, the validity of the research instrument was confirmed.

## 4. Results

As previously mentioned, the testing of the five hypotheses was performed by running paired *t*-test analyses. The results of the tests are illustrated in Tables 3 and 4.

**Table 4.** *t*-test results.

| | | Mean | Std. Deviation | Std. Error Mean | Lower | Upper | t | df | Sig. (2-Tailed) |
|---|---|---|---|---|---|---|---|---|---|
| | | **Paired Samples Test** | | | | | | | |
| | | **Paired Differences** | | | | | | | |
| | | | | | **95% Confidence Interval of the Difference** | | | | |
| Pair 1 | CULT_TRAD vs. CULT_VIR | 0.53285 | 1.21638 | 0.10392 | 0.32733 | 0.73836 | 5.127 | 136 | 0.000 |
| Pair 2 | LEAD_TRAD vs.LEAD_VIR | 0.62044 | 1.21839 | 0.10409 | 0.41459 | 0.82629 | 5.960 | 136 | 0.000 |
| Pair 3 | DIV_TRAD vs. DIV_VIR | 0.41119 | 1.57402 | 0.13448 | 0.14525 | 0.67713 | 3.058 | 136 | 0.003 |
| Pair 4 | COMM_TRAD vs. COMM_VIR | 0.61800 | 1.28738 | 0.10999 | 0.40050 | 0.83551 | 5.619 | 136 | 0.000 |
| Pair 5 | OBJ_TRAD vs.OBJ_VIR | 0.01168 | 1.28516 | 0.10980 | −0.20545 | 0.22881 | 0.106 | 136 | 0.915 |

The testing of the first hypothesis—H1: There was a significant difference between the perceptions of team culture strength in traditional versus virtual teams, showing a

significant difference between the means of the perceptions of team culture strength in traditional teams and the perceptions of team culture strength in virtual team (t(136) = 5.127, $p < 0.001$, N = 137). As indicated by the empirical data, the mean apposite for the perception of team culture strength in traditional teams (M = 4.0584, SD = 0.74049) was significantly greater than the mean apposite for the perception of team culture strength in virtual teams (M = 3.5255, SD = 0.87448), hence confirming H1.

Moving further, H2 presumed that there is a significant difference between the positive leadership perceptions in traditional versus virtual teams. The results of the *t*-test supported the inferred relationship (t(136) = 5.960, $p < 0.001$, N = 137), showing that the mean of the positive leadership perceptions in traditional teams (M = 4.1296, SD = 0.80217) was significantly greater than the mean of the positive leadership perceptions in virtual teams (M = 3.5091, SD = 0.85610).

The same situation applies to H3: there is a significant difference between team diversity approaches in traditional versus virtual teams (t(136) = 3.058, $p < 0.005$, N = 137). The findings reported that the mean of team diversity approaches in traditional teams (M = 3.7810, SD = 0.92570) was significantly greater than that specific to virtual teams (M = 3.3698, SD = 1.03349), thus confirming H3.

When it comes to communication effectiveness (i.e., H4: there is a significant difference between communication effectiveness in traditional versus virtual teams), the empirical data supported the presumed relationship (t(136) = 5.619, $p < 0.001$, N = 137), showing that the mean apposite for the communication effectiveness in traditional teams (M = 4.0730, SD = 0.77197) was significantly greater than that apposite for the communication effectiveness in virtual teams (M = 3.4550, SD = 0.88573). Consequently, H4 is supported.

H5 presumed that there is a significant difference between objectives achievement in traditional versus virtual teams. However, the results of the *t*-test did not support this assumption as $p > 0.05$, hence H5 was not confirmed in the context of the present research.

## 5. Discussion

Conflating the aforementioned results, four of the five research hypotheses were confirmed in the context of the current research. The findings revealed that the communicational and interactional processes within traditional teams (i.e., team members have face-to-face interactions) were valued more by the participants in the study than the corresponding ones taking place within virtual teams (i.e., team members have online interactions). Given the fact that all respondents had experienced both types of team activity, the obtained results could be considered as pertinent descriptors of the state-of-the-art for offline versus online interaction, communication, and collaboration among co-workers and between leaders and employees.

Respondents considered that the team culture is stronger built within traditional teams than in virtual teams. In this front, while working in traditional teams, they felt that their needs are better taken into consideration, that they are surrounded by people who share their values, that team spirit can be easily developed, that members are aware of the goals and objectives of the organization, and that face-to-face meetings are more important in order to create the culture of a team. These findings are consistent with previous studies [13,44,45] which have contended the difficulty of articulating team cohesion and team culture exclusively via online communication.

The same situation applies to the positive leadership perception within traditional teams versus virtual teams. Participants posited that traditional teams are more productive in this vein as the team leader succeeds in supporting teamwork, the communication on tasks between the team leader and the members of the team is more efficient, and the team leader directly inspires the team members while face-to-face communication with the leader allows a better clarification of vague issues. These aspects are in line with prior undertakings [13,27,51,56] which brought forward the special challenges encountered by leaders when trying to handle virtual teams.

When it comes to members' diversity, traditional teams offer more fertile ground for handling the reconciliation and alignment of team values and for surpassing intercultural differences, thus fostering a proper work climate for various members. These findings complement previous studies [13,59,60] which supported face-to-face interactions as prerequisites for intercultural adjustment and accommodation.

The analysis of communication effectiveness within traditional and virtual teams retrieved the same results in that respondents preferred face-to-face communication to online communication. Face-to-face communication allows them to work together efficiently for common achievements, to solve problems easier, to collaborate easier for tasks completion, to share the responsibilities for tasks easily, to achieve a better understanding of how objectives should be approached, and to ensure the clarity of task completion. The empirical evidence is in line with other previous findings [13,59,69] which have pointed to the potential disadvantages of virtual communication in contrast to face-to-face communication in relation to coordination and collaboration success.

No significant differences were retrieved between the achievement of objectives in traditional versus virtual teams. These results contrast previous studies [69] which revealed meaningful differences in this vein.

## 6. Conclusions

### 6.1. Summary of the Findings

The findings of the present study revealed that, from various points of view, teamwork in traditional teams (i.e., based on face-to-face interaction) is preferable to that in virtual teams (i.e., based on online interaction). The statistical analysis performed indicated that meaningful differences were observed regarding the perception of team culture strength, communication effectiveness, positive leadership perception, and diversity approach in favor of traditional teams, the questioned respondents opting for offline coordination and collaboration processes.

### 6.2. Research and Managerial Implications

The study has both research and managerial implications.

In terms of research implications, the current endeavor proposes an exploratory and comparative view on several main issues apposite for the work within virtual and traditional teams, with an emphasis on the new normal imposed by the COVID-19 pandemic. The study hence gives way to the problematization of the benefits and drawbacks on the catalysts and barriers which may emerge when dealing with hybrid work environments or when entirely substituting face-to-face interaction and communication with virtual collaboration. Even though there were no significant differences between the two types of teams in terms of objectives achievement, further research is expected to retest this particular issue in different settings.

In what concerns the managerial implications, this study may provide some valuable guidelines to leaders on the way they should approach work relationships within traditional and virtual teams. The advancement of the information and communication technologies has indeed facilitated better coordination among co-workers, yet it has fallen short to compensate for all the advantages of direct collaboration for the effectiveness of communication and strength of the team culture. In the light of the sustainable development objectives, leaders are dared to find feasible and proactive solutions to create a more productive work climate in the virtual environment on purpose to preserve and even increase the efficiency of the teamwork in the context of the COVID-19 pandemic.

Evidence was found supporting that, despite the presumed appetite for remote work which allows more flexibility and comfort in many cases, team members still attach greater importance to face-to-face interaction. This applies especially to the underlying team processes which account for the team cohesion and wellbeing, such as a strong team culture, efficient diversity management, consistent positive leadership perception, and communication effectiveness. Within this framework, reimagining work patterns and

dynamics emerges as an adaptive imperative while building hybrid environments may offer a reasonable compromise for all parties involved. In this way, leadership would benefit from a more articulate and positive perception, team members would benefit from knowing one another personally better, and a stronger team spirit and climate is liable to be objectivized by means of mixed interaction, all these ensuring a better team coordination for common objectives achievement. In the light of the current findings, the translation of teamwork exclusively into the virtual realms would afflicts teams at multiple levels, and therefore a moderate approach taking into consideration personal, social, and cultural-related factors would prove beneficial.

*6.3. Research Limitations and Future Directions*

The first research limitation which may be addressed via future studies refers to the number of the analyzed issues. Only five team-related aspects were specifically considered, namely the perception of team culture strength, positive leadership perception, diversity approach, communication effectiveness, and objectives achievement. Further research may expand the scope of the scrutiny to cover additional factors such as team dynamics, job satisfaction, team performance, etc. and thus address the shortcomings of the current exploration via a more comprehensive outlook of the team processes and patterns.

The second research limitation envisages the applied statistical tests which allow only an exploratory view on the investigated phenomena. The investigation did not study the underlying relationships among constructs, hence advancing only a preliminary overview on five key dimensions. Future studies may employ more complex methods and techniques for testing the relationships among the proposed factors within comparative models for traditional and virtual teams.

The third research limitation is related to the size of the convenience sample (N = 137), which impedes the generalization of the findings. Future studies may consider extending the number of the subjects or the targeted subjects by approaching respondents from more specific socio-demographic categories.

**Author Contributions:** Conceptualization, M.-E.S. and E.-M.V.; methodology, E.-M.V.; software, E.-M.V. and C.-M.V.; validation, L.-M.T., E.-M.V. and M.R.; formal analysis, E.-M.V.; investigation, M.-E.S.; resources, M.R.; data curation, L.-M.T.; writing—original draft preparation, M.-E.S.; writing—review and editing, E.-M.V.; visualization, C.-M.V.; supervision, E.-M.V.; project administration, E.-M.V.; funding acquisition, E.-M.V. All authors have read and agreed to the published version of the manuscript.

**Funding:** This work was supported by a grant of the Romanian Ministry of Education and Research, CNCS-UEFISCDI, project number PN-III-P1-1.1-TE-2019-1356, within PNCDI III.

**Institutional Review Board Statement:** Not applicable.

**Informed Consent Statement:** Informed consent was obtained from all subjects involved in the study.

**Data Availability Statement:** Not applicable.

**Conflicts of Interest:** The authors declare no conflict of interest.

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
