# Peer review of "Contrasting Traditional and Virtual Teams within the Context of COVID-19 Pandemic: From Team Culture towards Objectives Achievement"

_sustainability, doi:10.3390/su14084558_

Round 1

Reviewer 1 Report

Dear authors,

I appreciate your effort and I have some recommendations to enrich your manuscript:

  • The proposed method should be compared with the other state-of-the-art methods to demonstrate the efficiency of your method;
  • Pay attention to the accuracy of the data: in table 1 the percentage of female / male is not calculated correctly (81/137 = 59.12%). It would be preferable to check all the values so that there are no more errors;
  • The managerial implications identified are very general. It would be useful to identify some concrete implications generated by the research conducted;
  • Starting from the three limitations that were identified, the authors should describe how these limitations affect the results of their research.

Author Response

Dear Reviewer 1,

thank you for your kind suggestions and observations!

We have addressed all the issues you raised and we hope that the revised version of the article could be further considered for publication.

Best regards,

The authors

Reviewer 2 Report

The paper intends to compare some key dimensions of virtual and traditional teams. The dimensions considered in the paper are: leadership perception, communication among team members, objectives achievement, diversity approach, and team culture. Five hypotheses are stated and then tested, based on the questionnaire data collected.
The paper is clearly written, the structure of the paper is accurate (besides a few minor comments that can be made to the "literature review section"), the results are quite well documented and supported by appropriate tables, the conclusions are legitimate. In summary: the article is a good read. However, there are some important elements missing in the text and some major things that need to be corrected.

List of major changes:

1) The sustainability context is completely missing in the text. The purpose of the study need to be clearly linked to and explained with reference to the sustainability dimensions.

2) The literature review takes up a good portion of the text and has been carefully conducted, however, it lacks a clear summary of previous (similar) research and highlights what new research brings in comparison to the existing literature. It is only in the Discussion section (lines: 525, 534, 539, 547, 551) that references to similar articles appear, but they also lack highlighting what was missing in those studies, which is completed in this article. Ideally, a description of the research gap would have been included in the literature review section.

3) The proposed hypotheses, as formulated (Lines: 198, 259, 302, 361, 426 and also: Table 2), appear to be unmeasurable. The question to be verified is whether good names have been used for the hypotheses adopted? Is what is being tested the culture itself (unmeasurable) or the strength with which the team's culture affects the team (measurable). A similar observation applies to communication patterns (unmeasurable). Perhaps it was the effectiveness of communication that was measured? Also in the case of leadership: the question is whether what was being tested was not leadership, but the strength of leadership?
  4) Some important statements about the research conducted require further explanation:
- Can at least one literature reference to the "snowball sampling technique" that was used for data collection be inserted? The technique is being mentioned in line 431.
- Line 455 refers to "Twelve main conceptual dimensions", but what are these dimensions (they are not listed anywhere)?
- Line 458 refers to "previous studies", but the literature reference is also missing here.
- Line 463 mentions Cronbach’s alpha, but it does not detail how it was calculated (having the given set of data that was collected).
- Line 464 focuses on "values above the threshold of 0.7". Is this threshold somehow justified? If so, can it be referenced to the literature?
- What is the scale of responses used in the questionnaires? (Table 2)

List of minor changes:
5) The literature review section would benefit from subsections corresponding to the hypotheses discussed in turn.

6) The items being referenced are not always fully described: e.g., Line 50 mentions "more substantial changes", but can they be listed? Line 122 should not only mention but also list the "challenges never experienced until now". A similar comment applies to line 139 and opportunities.

7) Some minor corrections are needed: Line 331 states "Krawczyk-Bryłka show" should be changed to "Krawczyk-Bryłka shows" (as this is one author only). Line 477 refers to "six hypotheses", but there are only five hypotheses (not six) that were tested. Line 454 states that "52 statements were designed as paired items", but Table 2 lists only 46 items (5 + 4 + 3 + 6 + 5) * 2. Is it a mistake or some items were not shown in Table 2?

Author Response

Dear Reviewer 2,

thank you for your kind suggestions and thorough observations!

We have addressed all the issues you raised and we hope that the revised version of the article could be further considered for publication.

Best regards,

The authors

Reviewer 3 Report

I have only two key comments in the amendments; I recommend:
1. to extend the abstract, as stated in the guidelines for authors, in particular by the scientific methods used and the results of the authors' research,
2. it is appropriate to extend the scientific resources used and the theoretical basis by
 authors' ideas that directly or indirectly relate to the topic under study, such as:

Peracek, T. HUMAN RESOURCES AND THEIR REMUNERATION: MANAGERIAL AND LEGAL BACKGROUND. 13th International Scientific Conference on Reproduction of Human Capital - Mutual Links and Connection (RELIK) 2020: REPRODUCTION OF HUMAN CAPITAL - MUTUAL LINKS AND CONNECTIONS, pp.454-465

Adamišin, P., Šindleryová, I.B., Čajková, A .(2021). CORONAVIRUS VS. REAL CAUSE OF THE EUROPEAN ECONOMIC CRISIS - COMPARING SLOVAK AND GERMAN NATIONAL MODEL EXAMPLE. Online Journal Modeling the New Europe, 2021, (37), pp. 78–101, doi: 10.24193 / OJMNE.2021.37.05

Skrabka, J. The moratorium on loan repayments during the Covid-19 Pandemic in Europe: a comparative analysis of loan moratoria in selected European countries. 2021, JURIDICAL TRIBUNE-TRIBUNA JURIDICA, 11 (SI), pp.291-301, doi: 10.24818 / TBJ / 2021/11 / SP / 02

Mucha, B. (2021). Evaluation of the State of Implementation of the European Structural and Investment Funds: Case Study of the Slovak Republic. Online Journal Modeling the New Europe, 35, 4-24, doi: 10.24193 / OJMNE.2021.35.0

Plavčan, P. & Funta, R. (2020). Some Economic Characteristics of Internet Platforms. Danube, 11 (2), 156–167, doi: 10.2478 / danb-2020-0009

Srebalová, M. & Vojtech, F. (2021). SME Development in the Visegrad Area. Eurasian Studies in Business and Economics, 17, pp. 269–281, doi: 10.1007 / 978-3-030-65147-3_19

Author Response

Dear Reviewer 3,

thank you for your kind suggestions and observations!

We have addressed all the issues you raised and we hope that the revised version of the article could be further considered for publication.

Best regards,

The authors

Round 2

Reviewer 2 Report

Thank you for taking into account my comments. I accept the article in present form.